# Improving Tool-Using Language Agents via MDL-Guided Rule Learning

## Abstract

Large language models (LLMs) often struggle to use tools reliably in domain-specific settings, where APIs may be idiosyncratic, under-documented, or tailored to private workflows. This highlights the need for effective adaptation to task-specific tools. We propose RimRule, a neuro-symbolic approach for LLM adaptation based on dynamic rule injection. Compact, interpretable rules are distilled from failure traces and injected into the prompt during inference to improve task performance. These rules are proposed by the LLM itself and consolidated using a Minimum Description Length (MDL) objective that favors generality and conciseness. Each rule is stored in both natural language and a structured symbolic form, supporting efficient retrieval at inference time. Experiments on tool-use benchmarks show that this approach improves accuracy on both seen and unseen tools without modifying LLM weights. It outperforms prompting-based adaptation methods and complements finetuning. Moreover, rules learned from one LLM can be reused to improve others, including long reasoning LLMs, highlighting the portability of symbolic knowledge across architectures.

## 1 Introduction

Humans adapt by trying, erring, and compressing experience into reusable heuristics (Kolb, 1984; Metcalfe, 2017; Gigerenzer & Gaissmaier, 2011). Such compact representation of regularities achieve cognitive economy—they preserve what matters for performance while discarding incidental detail, making knowledge easier to reuse across tasks and by other people (Rosch, 1978).

Large language models (LLMs) (Brown, 2020; Bai et al., 2023; DeepMind, 2024; Grattafiori et al., 2024) also require adaptation, especially when deployed in unfamiliar domains with specialized tools or under-documented APIs. However, the way LLMs adapt differs markedly from how humans do. Today, adaptation typically takes one of three forms: (i) retrieved examples for few-shot prompting (Brown et al., 2020; Liu et al., 2021), (ii) globally tuned prompts (Pryzant et al., 2023; Cui et al., 2025), or (iii) fine-tuned model weights (Hu et al., 2022). Each approach is powerful, but none supports abstraction, reuse, and interpretability in the way humans generalize from experience. Few-shot prompting reuses raw supervision but does not abstract—and LLMs often extract only shallow patterns from examples (Wei et al., 2023); global prompts are static and brittle in interactive settings (Verma et al., 2024); and tuning model weights is costly, requiring retraining whenever the environment changes, and preventing easy sharing of acquired knowledge across models (Yang et al., 2024).

We explore a fourth paradigm for adapting LLMs—one that mirrors how humans learn from failure. Instead of tuning weights or retrieving demonstrations, we induce interpretable rules. At training time, these rules are proposed in response to observed failures. At inference time, relevant rules are dynamically retrieved and injected into the prompt to improve LLM performance. This constitutes a distinct adaptation paradigm: unlike few-shot prompting, it emphasizes abstraction and compression over raw example replay; unlike finetuning, it produces symbolic, interpretable artifacts. Because these rules are both understandable to humans and legible to modern LLMs, they can be reused across different LLMs without retraining.

A central challenge, however, is *consolidation*. Humans do not retain every rule or exception encountered; instead, they refine their knowledge by merging, pruning, and updating rules into a more compact and general set. This process supports cognitive economy and guards against overload

(Rosch, 1978). We formalize this process using the *Minimum Description Length* (MDL) principle, which views good rules as those that compress experience by balancing complexity and explanatory power (Rissanen, 1978a; Grünwald, 2007). In our formulation, rules that are overly specific or redundant are merged or removed to minimize the total description length of both the rule library and the residual interaction traces. This yields a high-coverage, low-redundancy library that generalizes beyond individual examples.

Another key challenge is *scalability*: learning regularities across large amount of training samples requires efficient generation and consolidation. To address this, we leverage a *distributed rule generation* process, in which candidate rules are proposed in parallel from independent failures, enabling scalable rule mining across interaction logs. In addition, each rule is stored in a dual format: a human-readable natural language version and a domain-specific symbolic representation. The latter not only improves consolidation, by enabling structural matching and overlap detection, but also makes rule retrieval efficient during inference.

We demonstrate this approach on two tool-use benchmarks—ToolHop (Ye et al., 2025) and BFCL (Yan et al., 2024)—where LLMs must reason over tool descriptions and invoke them correctly. Simple retry mechanisms based on tool error messages, or even the use of advanced long reasoning models, often fall short in reliably correcting systematic failures. Our results highlight several key observations.

- First, inference-time rule injection consistently improves performance, including on queries involving tools that never appear in training.

- Second, learned rules transfer across LLMs: rules distilled from a smaller or less capable LLM can improve the performance of stronger, long-reasoning models.

- Third, our method outperforms prompting-based adaptation approaches and complements finetuning, offering a distinct and synergistic axis of generalization.

## 2 METHOD

We introduce RIMRULE[1], a two-stage, scalable learning framework that equips LLM-based agents with inference-time rules distilled from failure traces. Instead of fine-tuning model parameters, RIMRULE proposes interpretable rules that capture the regularities underlying past errors and uses them to guide future decisions at inference time.

### 2.1 A TWO-STAGE APPROACH

Figure 1: The framework of our MDL-guided rule learning.

---

[1] **R**eusable, **I**nterpretable, and **M**DL-guided Rules

Sequential rule learning, where failures are processed one by one and rules are committed immediately, suffers from path dependence and poor scalability. In separate-and-conquer rule induction (Clark & Niblett, 1989; Cohen, 1995b; Fürnkranz, 1999), early rules reshape the remaining data, often leading to over-specific or brittle patterns. This sensitivity to order motivates ensembling (Breiman, 1996), and is further amplified in our setting, where data are noisy and rules are expressed in natural language.

Could randomizing the data order help? For neural models trained with SGD, reshuffling batches yields approximately unbiased updates, and order mainly affects variance (Robbins & Monro, 1951; Bottou, 2010; Hardt et al., 2016). But rule induction is a discrete, greedy process: once a rule is added, the residual distribution shifts, and subsequent proposals change. There is no averaging across orders—just different trajectories through the rule space.

Scalability is another limitation. Sequential learning precludes parallelism and scales poorly with data volume. In contrast, distributed pattern extraction is a natural fit for our setting (Dean & Ghemawat, 2004; Agrawal & Shafer, 1996).

Motivated by these challenges, RimRule adopts a *two-stage* architecture (Figure 1). First, it performs *distributed rule generation*: each failure case is processed independently, generating candidate rules in parallel. Second, it performs *rule consolidation*, resolving redundancy and selecting a compact library under an MDL objective. The next two sections describe each stage in detail, and a pseudocode implementation is included in Appendix A.1.

## 2.2 Distributed Rule Generation

This stage has two components. First, we perform *local rule generation*, where rules are induced directly from failures by comparing incorrect and ground-truth traces. Each failure case is processed independently, allowing rule proposals to be generated in parallel across the dataset—eliminating order effects and improving scalability. Second, we translate the natural-language rules into a compact *symbolic representation*, which enables principled consolidation under MDL and efficient retrieval at inference time.

### 2.2.1 Local Rule Generation

We assume access to *ground-truth execution traces* during training, as well as a scalar *performance metric* $m(\tau) \in [0, 1]$ that measures the quality of an execution trace $\tau$.[2] A trace is considered a failure if it receives a score $m(\tau^-) < 1$.

Let each training sample be a tuple $(x, \mathcal{S}, \tau^-, \tau^*)$, where: $x \in \mathcal{X}$ is the user query, $\mathcal{S}$ is the set of available tools, $\tau^-$ is the agent's incorrect execution trace, and $\tau^*$ is the corresponding ground-truth trace. Incorrect traces may be *complete* (returning an incorrect answer) or *incomplete* due to execution errors. The generator compares $\tau^-$ and $\tau^*$, identifies a root-cause *reasoning* failure (as opposed to downstream propagation), and proposes a compact rule $r \in \mathcal{R}_0$ that corrects or prevents the error on this and similar examples. Each rule $r$ is tagged with an error type $d(r) \in \{\text{dec}, \text{sel}, \text{arg}\}$, corresponding to decomposition, tool selection, or argument construction, respectively.

To convert a concrete failure into an abstract, reusable rule, we follow the template of *Explanation-Based Learning* (EBL): begin with generating a grounded, detailed explanation, then generalize by abstracting away instance-specific details (Mitchell et al., 1986; DeJong & Mooney, 1986). The rule generator is instantiated as an LLM-based function.[3]

Each proposed rule is evaluated using two automatic filters. First, a *predictive check*: we inject the rule back into the agent, re-run it on the same query $x$, and check whether the resulting trace $\hat{\tau}$ improves the performance score $m(\hat{\tau})$. Second, a *linguistic check*: the rule must satisfy format constraints such as a clean *if-then* structure and bounded length. Rules failing either check are discarded or regenerated.

---

[2]Ground-truth traces are used only at training time to localize errors. In settings without traces but with a reward function or score, the same mechanism can be adapted by proposing multiple candidate rules and selecting those that maximize observed reward on the instance; a full treatment is left to future work.

[3]See appendix for the prompt.

Since a single rule often fixes only one error in the trace, we allow generating multiple rules per training sample. After proposing a rule and observing an improvement (i.e., $m(\hat{\tau}) > m(\tau^-)$) but not a full correction (i.e., $m(\hat{\tau}) < 1$), we re-invoke the generator to propose an additional rule, conditioned on the residual error. This process continues iteratively until the trace is fully corrected or no further atomic fixes can be proposed.

### 2.2.2 SYMBOLIC REPRESENTATION

Natural-language renderings of the same rule often differ in phrasing, which obstructs both structural matching and consistent description-length accounting. To address this, we compile each rule into a symbolic form: a structured representation with fixed fields and closed vocabularies. This enables well-defined description length computation (Section 2.4) and efficient retrieval (Section 2.3).

The symbolic schema defines five semantic fields $\mathcal{F}$: `Domain` (broad topic), `Qualifier` (situation or context), `Action` (prescribed operation), `Strength` (e.g., always or consider), and `ToolCategory` (abstract tool type). These fields reflect common axes of reasoning in structured agent behavior (Wang et al., 2017; Feng et al., 2018; Tambwekar et al., 2021). Each field $f \in \mathcal{F}$ has an associated vocabulary $\mathcal{V}_f$, and the overall vocabulary is $\mathcal{V} = \bigcup_{f \in \mathcal{F}} \mathcal{V}_f$.

We construct $\mathcal{V}$ by prompting an LLM to suggest candidate tokens for each field, using batches of natural-language rules $\{r_i^{\text{NL}}\}$ as input. To scale, we apply this procedure iteratively across batches, each time presenting the current vocabulary so that the model can reuse or extend it. As this process introduces mild order dependence, we repeat it across randomized orderings and select the vocabulary yielding the smallest overall size. This favors compact, semantically consistent vocabularies without requiring domain-specific heuristics.

Once the vocabulary is frozen, we translate each rule $r_i^{\text{NL}}$ into its symbolic representation $r_i^{\text{sym}} \in \mathcal{R}^{\text{sym}}$ by prompting an LLM to assign valid field values from $\mathcal{V}$. This yields a deterministic, auditable rule set that supports downstream MDL-guided consolidation and inference-time retrieval.

In addition to the semantic fields $\mathcal{F}$, each rule $r \in \mathcal{R}^{\text{sym}}$ also carries auxiliary metadata. This includes the error type $d(r) \in \{\text{dec}, \text{sel}, \text{arg}\}$, indicating the reasoning dimension the rule addresses; a scope flag $\sigma(r) \in \{\text{tool-bound}, \text{category}\}$, which specifies whether the rule applies to a specific tool or a broader category (Section 2.4); and, for tool-use rules, a provenance field indicating the original tool associated with the rule at generation time.

### 2.3 RULE RETRIEVAL

The symbolic structure of rules enables retrieval that is both efficient and interpretable. Compared to alternatives such as raw semantic similarity over natural language or end-to-end retrieval via large language models, symbolic matching is cheaper, more scalable, and more robust to incidental lexical variation (Tan et al., 2023; Liu et al., 2023). Semantic similarity can be diluted by non-essential content, while LLM-based rule selection becomes prohibitively expensive as the rule library grows. By structuring both queries and rules into a shared symbolic format, we can perform accurate, low-overhead matching based on meaningful fields.

At inference time, given a query $x_q$ and its available tools $\mathcal{S}_q$, we first convert the current state into a symbolic representation $z_q$, structured over the symbolic fields $\mathcal{F}$. Retrieval then proceeds in two stages: a coarse filter to discard inapplicable rules, followed by a fine-grained ranking step over the remaining candidates.

During coarse filtering, all rules with error type $d(r) = \text{dec}$ (decomposition) are retained unconditionally. For tool-use rules ($d(r) \in \{\text{sel}, \text{arg}\}$), applicability is determined by a scope flag $\sigma(r) \in \{\text{tool-bound}, \text{category}\}$. If $\sigma(r) = \text{tool-bound}$, the rule is retained only if its bound tool name appears in $\mathcal{S}_q$. If $\sigma(r) = \text{category}$, the rule is retained if its `ToolCategory` matches that of any tool in $\mathcal{S}_q$. This mechanism allows tool-specific rules to be generalized, during consolidation (Section 2.4), to apply more broadly to entire categories of tools.

After filtering, we apply a ranking function to prioritize the remaining rules. For each rule $r$, we compute the average semantic similarity between its symbolic fields and those of the query $z_q$, using an embedding model such as Sentence-BERT (Reimers & Gurevych, 2019). Rules are ranked by this score, and we retain the top $k$ candidates. Each selected rule is injected into the agent's context

in its natural language form, which is more interpretable to the LLM and better suited for guiding reasoning.

## 2.4 MDL-Guided Rule Consolidation

Unlike neural networks, where the model size is fixed and optimization focuses solely on parameters, our learned adaptation is a symbolic rule library whose size and content evolve during training. Rules can be added, removed, or generalized, and each change affects both the expressiveness and complexity of the system. To balance this trade-off in a principled way, we adopt the Minimum Description Length (MDL) principle: a classic formalism that favors models which compress the observed data well while remaining concise themselves (Rissanen, 1978b; Grünwald, 2007). MDL naturally penalizes redundancy and overfitting, encourages generalization when it yields consistent gains, and provides a unified cost function for optimizing both what the rule library says and how well it works.

Let $H \subseteq \mathcal{R}^{\text{sym}}$ denote a proposed rule subset to be retained. We aim to select a compact, high-impact library $H$ by minimizing the total description length:

$$\text{MDL}(H) = L(H) + L(D \mid H),$$

where $L(H)$ is the model cost of encoding the rules and $L(D \mid H)$ is the cost of encoding the observed agent behavior given those rules.

We define the model prior $P(H)$ via a length-based Gibbs distribution over symbolic rule sets:

$$P(H) \propto \exp\Big(-\alpha \sum_{r \in H} \ell(r)\Big),$$

where $\ell(r)$ is the token length of rule $r$, and $\alpha > 0$ is a regularization strength. This is the maximum-entropy distribution under a constraint on expected aggregate rule length (Jaynes, 1957), and is consistent with the coding perspective of Shannon and Kraft (Shannon, 1948; Kraft, 1949; McMillan, 1953). The corresponding code length is

$$L(H) = -\log P(H) = \alpha \sum_{r \in H} \ell(r) + \log Z(\alpha),$$

where $Z(\alpha)$ normalizes over all subsets of $\mathcal{R}^{\text{sym}}$. Since $Z(\alpha)$ is constant for a fixed candidate pool, it is dropped during optimization.

Let $D = \{(x_i, \mathcal{S}_i, \tau_i^-, \tau_i^*)\}_{i=1}^n$ be the training set, and for each rule set $H$, let $a_i(H) \in \{0, 1\}$ indicate whether the failure in sample $i$ is corrected after injecting $H$. Denote $k_H = \sum_{i=1}^n a_i(H)$ and $\hat{p}_H = k_H/n$. The plug-in refined-MDL codelength under a Bernoulli likelihood is:

$$L(D \mid H) = -\Big[k_H \log \hat{p}_H + (n - k_H) \log(1 - \hat{p}_H)\Big] + \log C_n,$$

where $C_n$ is the normalizing constant for the Bernoulli NML universal code (Shtarkov, 1987; Grünwald, 2007). Since $\log C_n$ is constant across hypotheses, we minimize the negative log-likelihood term.

We minimize $\text{MDL}(H)$ by starting from the full rule pool $H_0 = \mathcal{R}^{\text{sym}}$ and applying local edits that strictly reduce the objective. Let $\Delta_{\text{data}} = L(D \mid H') - L(D \mid H)$ denote the change in data cost under a proposed modification $H \to H'$. We consider two types of edits:

- *Prune:* Remove a rule $r \in H$. Accept the edit $H' = H \setminus \{r\}$ if
$$\Delta_{\text{prune}} = -\alpha\,\ell(r) + \Delta_{\text{data}} < 0.$$

- *Generalize:* Flip the scope flag of a rule $r$ from $\sigma(r) = \text{tool-bound}$ to $\sigma(r^{\text{gen}}) = \text{category}$, creating a new rule $r^{\text{gen}}$ with reduced specificity and shorter token length. Accept the edit $H' = (H \setminus \{r\}) \cup \{r^{\text{gen}}\}$ if
$$\Delta_{\text{gen}} = \alpha\,[\ell(r^{\text{gen}}) - \ell(r)] + \Delta_{\text{data}} < 0.$$

We apply these edits in a greedy fashion: for each rule in the current library, we evaluate potential prune and generalize operations, accepting only the one that yields the greatest reduction in the MDL objective. We repeat this process until the objective no longer improves.

## 3 EXPERIMENTS

### 3.1 DATASETS

We evaluate our system on two tool-use benchmarks: **ToolHop** Ye et al. (2025) and **BFCL** Yan et al. (2024). ToolHop features compositional, multi-turn queries that require chaining multiple tools across reasoning steps. To provide a complementary setting, we use the `live-multiple` subset of BFCL, which, despite being single-step, presents more challenging tool selection due to a larger and more diverse tool set.

For both datasets, we assume access to ground-truth execution traces in the training data, but no ground-truth rules are provided.

To assess generalization, we define two evaluation splits for each dataset: - **test-rand**: A random split of queries used to evaluate in-distribution performance. - **test-unseen**: A held-out split constructed from queries involving tools not seen during training, selected based on tool rarity.

Table 6 summarizes the number of examples per split. We use **test-unseen** to evaluate generalization to novel tools, and **test-rand** to measure improvement under distributional overlap.

### 3.2 REFERENCE METHODS

All methods in our evaluation prompt LLMs to perform tool-augmented reasoning using a **ReAct**-style format (Yao et al., 2022), where the model is asked to generate sub-steps and tool calls interactively. We allow the LLM agent to **retry** based on observed tool feedback or error messages—a mechanism that improves robustness but often fails to recover from systematic reasoning errors. Our method builds on top of this retry setup by injecting explicit rules to reduce failure in the first place.

We compare RIMRULE against several established adaptation paradigms. These methods serve as reference points for evaluating both standalone effectiveness and complementarity when combined with our approach.

**SEE** (Cui et al., 2025) is an automatic prompt-optimization framework that jointly evolves instructions and in-context examples using LLM-driven evolutionary strategies. It treats the prompt as a single global object and optimizes it holistically. In contrast, our method adapts *per instance*, learning from failures and accumulating reusable rules that are retrieved step-wise rather than injected as a monolithic prompt.

**Few-shot In-Context Learning** selects the top-$k$ training examples as demonstrations. Relevance is computed using a weighted combination of semantic similarity between test and training queries (via Sentence-BERT) and overlap in tool availability (based on tool descriptions). The top-$k$ examples are inserted as in-context demonstrations.

**Supervised finetuning (SFT)** is implemented using LoRA (Hu et al., 2022). We train models with standard supervised objectives on the tool-use tasks. We also evaluate foundation models that are pre-finetuned for function-calling capabilities.

**These reference methods use the same training set as ours** and differ only in how they encode adaptation signals. We evaluate both standalone and combined versions, demonstrating that RIM-RULE provides additive gains when layered on top of prompting, SFT, or long-reasoning baselines.

To test whether strong reasoning capabilities alone can resolve tool-use errors, we also include **long-reasoning models** (e.g., `o1`) in our evaluation. While these models perform well under zero-shot prompting, we find that they still benefit from rule-based guidance—highlighting the complementary role of symbolic rules even in high-capacity systems.

### 3.3 IMPLEMENTATION DETAILS

All experiments are conducted using publicly available open-weight models and APIs. Appendix details the prompts ( A.3) used for rule generation and canonicalization, dataset statistics and LLM checkpoints ( A.2).

Table 1: Accuracy ($\pm$ standard deviation, in %) and number of rules at each learning stage. Rule consolidation improves performance while reducing rule count.

| | ToolHop | | | BFCL | | |
|---|---|---|---|---|---|---|
| | Test-rand | Test-unseen | # Rules | Test-rand | Test-unseen | # Rules |
| Initial | $26.5_{\pm1.3}$ | $35.1_{\pm1.6}$ | 0 | $50.1_{\pm0.9}$ | $45.0_{\pm1.0}$ | 0 |
| + Rule Gen. | $27.6_{\pm1.3}$ | $42.1_{\pm1.6}$ | 72 | $54.8_{\pm1.3}$ | $47.6_{\pm1.4}$ | 151 |
| + Consolid. | $\mathbf{31.1}_{\pm1.3}$ | $\mathbf{43.1}_{\pm1.6}$ | 67 | $\mathbf{56.6}_{\pm1.2}$ | $\mathbf{48.5}_{\pm1.4}$ | 121 |

## 4 RESULTS AND DISCUSSION

### 4.1 LEARNING PROCESS

We illustrate the learning trajectory in Table 1. Initially, the rule library is empty. Using Llama3.2 as the tool agent, we collect failure traces across training queries. During the distributed rule generation stage, we generate 72 rules for ToolHop and 151 rules for BFCL.

This initial rule set improves performance on both evaluation splits. Accuracy increases on the **test-rand** split (which shares tools and distributional patterns with training) as well as the more challenging **test-unseen** split (which contains only tools not seen during training). A sample of learned rules are provided in Appendix A.4.

Next, we apply MDL-guided consolidation to prune redundant rules and generalize overly specific ones. This reduces the number of rules and further improves performance on **test-rand**. Gains on **test-unseen** are smaller but still positive—likely because the MDL objective is optimized over the training distribution, which better matches **test-rand** than **test-unseen**.

Overall, the learning process yields a compact, interpretable rule library that improves both in-distribution and out-of-distribution generalization.

### 4.2 RE-USABILITY ACROSS LLMS

Humans can share and reuse heuristics because they are interpretable—not just effective in context, but understandable across settings. By encoding adaptation in a symbolic, human-readable form, our method enables rules to be reused across different models with no retraining.

We evaluate whether the rules learned from one LLM can be reused to improve others. Specifically, we train two rule libraries independently: one from failure traces produced by **Llama3.2**, and another from **GPT-4o**. We then apply both sets of rules across a suite of models with varying architecture, size, and reasoning strength, and measure performance on the **test-rand** split.

Table 2 shows that both rule libraries consistently improve performance across models, regardless of which model they were learned from. Notably, even strong reasoning models such as **O1** and **Llama4** still benefit from rule injection, suggesting that symbolic rules capture high-level failure patterns that are not automatically addressed by scale or pretraining. Moreover, rules are not tightly coupled to the source model—they generalize well across LLMs, enabling reuse without retraining.

### 4.3 PERFORMANCE ON SMALL DATASET

We evaluate the effectiveness of RIMRULE in low-resource settings by training on a reduced dataset. Specifically, we experiment on the multi_turn_base split of BFCL using only 90 training samples (Table 6). Despite the limited supervision, RIMRULE successfully learns 4 rules (Appendix A.4), which are then used at inference time. These rules yield substantial improvements: accuracy on **test-rand** increases from 55.2% to 62.1%, and accuracy on **test-unseen** improves from 46.0% to 60.0%. These results suggest that even a small number of well-targeted rules can provide meaningful gains—highlighting the sample efficiency and practical applicability of our approach.

Table 2: Accuracy ($\pm$ standard deviation) of LLMs on Toolhop and BFCL, with and without rules. Rules are learned once from Llama3.2 and reused across models.

| Learned from | Applied on | ToolHop | | BFCL | |
|---|---|---|---|---|---|
| | | No Rules | RimRule | No Rules | RimRule |
| Llama3.2 | Llama3.2 | $26.5_{\pm1.3}$ | $\mathbf{31.1}_{\pm1.3}$ | $50.1_{\pm0.9}$ | $\mathbf{56.6}_{\pm1.2}$ |
| | GPT-4o | $58.1_{\pm1.4}$ | $57.4_{\pm1.4}$ | $71.6_{\pm0.8}$ | $\mathbf{75.6}_{\pm1.1}$ |
| | Llama4 | $73.8_{\pm1.2}$ | $\mathbf{76.7}_{\pm1.2}$ | $75.8_{\pm0.8}$ | $\mathbf{77.5}_{\pm1.1}$ |
| | O1 | $53.2_{\pm1.4}$ | $\mathbf{57.4}_{\pm1.4}$ | $75.6_{\pm0.8}$ | $\mathbf{77.6}_{\pm1.1}$ |
| GPT-4o | Llama3.2 | $26.5_{\pm1.3}$ | $\mathbf{31.3}_{\pm1.3}$ | $50.1_{\pm0.9}$ | $\mathbf{52.4}_{\pm1.3}$ |
| | GPT-4o | $58.1_{\pm1.4}$ | $\mathbf{60.3}_{\pm1.4}$ | $71.6_{\pm0.8}$ | $\mathbf{76.4}_{\pm1.1}$ |
| | Llama4 | $73.8_{\pm1.2}$ | $\mathbf{77.4}_{\pm1.2}$ | $75.8_{\pm0.8}$ | $\mathbf{77.7}_{\pm1.0}$ |
| | O1 | $53.2_{\pm1.4}$ | $\mathbf{56.1}_{\pm1.2}$ | $75.6_{\pm0.8}$ | $\mathbf{77.9}_{\pm1.1}$ |

Table 3: Comparison of prompting-based adaptation methods on ToolHop and BFCL.

| | ToolHop | | BFCL | |
|---|---|---|---|---|
| | Test-rand | Test-unseen | Test-rand | Test-unseen |
| Zero-shot (Yao et al., 2022) | $26.5_{\pm1.3}$ | $35.1_{\pm1.6}$ | $50.1_{\pm0.9}$ | $45.0_{\pm1.0}$ |
| Few-shot (Liu et al., 2021) | $29.9_{\pm1.4}$ | $37.9_{\pm1.4}$ | $54.5_{\pm0.9}$ | $46.6_{\pm1.4}$ |
| SEE (Cui et al., 2025) | $27.6_{\pm1.5}$ | $35.9_{\pm1.5}$ | $52.2_{\pm1.0}$ | $45.5_{\pm1.0}$ |
| RimRule (Ours) | $\mathbf{31.1}_{\pm1.3}$ | $\mathbf{43.1}_{\pm1.6}$ | $\mathbf{56.6}_{\pm1.2}$ | $\mathbf{48.5}_{\pm1.4}$ |

## 4.4 COMPARISON WITH PROMPTING-BASED METHODS

We first compare RimRule to prompting-based adaptation strategies, including zero-shot prompting (Yao et al., 2022), few-shot in-context learning (Liu et al., 2021), and prompt optimization via SEE (Cui et al., 2025). All methods operate over the same training data and tool descriptions, and differ only in how they encode task-specific supervision.

Table 3 shows that RimRule consistently outperforms all three prompting strategies across both ToolHop and BFCL datasets, on both **test-rand** and **test-unseen** splits. While SEE and few-shot ICL offer modest improvements over zero-shot performance, their gains plateau and do not generalize well. In contrast, our rule-based framework yields larger improvements and stronger out-of-distribution generalization, highlighting the value of structured, reusable inference-time regularities over fixed prompts.

## 4.5 COMPLEMENTING FINETUNED MODELS

While finetuning can substantially improve LLM performance on tool-use tasks, it tightly couples adaptation to the model's internal weights and often struggles to generalize beyond the training distribution. In contrast, RimRule provides an interpretable and modular adaptation layer that can be composed with finetuning without interfering with its core capabilities.

As shown in Table 4, applying RimRule on top of finetuned models leads to consistent performance improvements—particularly on the **test-unseen** split. This suggests that symbolic rules help correct residual, systematic reasoning errors that remain even after supervised training. The gains highlight that rule-based adaptation offers a complementary, inference-time mechanism for improving LLM behavior.

Modern LLMs are increasingly finetuned with native function calling (FC) capabilities during post-training, allowing them to execute structured API-like queries. To test whether these models still benefit from explicit symbolic rules, we evaluate RimRule on top of function-calling–enabled LLMs. As shown in Table 5, even LLMs that have been specially finetuned for tool use continue to benefit from the addition of symbolic rules. These results further support the claim that RimRule

Table 4: Performance of **Llama3.2** with and without RIMRULE under supervised finetuning (SFT).

| | ToolHop | | BFCL | |
|---|---|---|---|---|
| | Test-rand | Test-unseen | Test-rand | Test-unseen |
| SFT | $43.8_{\pm 1.4}$ | $38.5_{\pm 1.5}$ | $65.0_{\pm 0.9}$ | $58.7_{\pm 1.4}$ |
| + RIMRULE | $\mathbf{50.0}_{\pm 1.4}$ | $\mathbf{45.1}_{\pm 1.6}$ | $\mathbf{68.6}_{\pm 1.2}$ | $\mathbf{62.7}_{\pm 1.3}$ |

Table 5: Performance of **GPT-4o** with and without RIMRULE under function-calling prompting.

| | ToolHop | | BFCL | |
|---|---|---|---|---|
| | Test-rand | Test-unseen | Test-rand | Test-unseen |
| Function Calling | $63.8_{\pm 1.4}$ | $83.0_{\pm 1.2}$ | $79.6_{\pm 0.7}$ | $77.2_{\pm 0.8}$ |
| + RIMRULE | $\mathbf{66.1}_{\pm 1.4}$ | $\mathbf{85.4}_{\pm 1.1}$ | $\mathbf{81.7}_{\pm 1.0}$ | $\mathbf{79.8}_{\pm 1.0}$ |

operates along an orthogonal axis of generalization—complementing both task-specific finetuning and architectural improvements.

## 5 RELATED WORK

Early symbolic systems—from decision trees like ID3 Quinlan (1986) and rule lists like RIPPER Cohen (1995a), to logical learners like FOIL Quinlan (1990)—offered interpretable, modular reasoning. Rule pruning, both at the literal and rule-set level, emerged as a key strategy for improving generalization Quinlan (1987); Fürnkranz (1997), with the MDL principle providing a unifying foundation Grunwald (2007). While these systems targeted structured, tabular domains, they inform our approach to consolidating symbolic rules in LLM-driven environments.

Recent work has revisited rule learning in the LLM setting. In prompting-based methods, rules are extracted from demonstrations Gao & Das (2024), graphs Chen et al. (2024), or offline traces Zhang et al. (2024), and used to guide generation Zhou et al. (2024); Wang et al. (2024b). In training-time approaches, rules supervise synthetic data Morishita et al. (2024), reward functions Wang & Xiong (2025), or distillation Sadeq et al. (2025). Some systems store symbolic rules in memory Wang et al. (2024a), but most prior work learns rules statically, with limited feedback or reuse across tasks.

## 6 CONCLUSION

We present a lightweight yet principled approach for adapting large language models to new domains by distilling compact, interpretable rules from failure traces. These rules serve as an inference-time scaffold—learned without modifying model weights, stored in a dual-format symbolic structure, and consolidated via MDL to promote generalization and reuse. Our method improves both in-distribution and out-of-distribution performance, outperforms prompting-based alternatives, and complements finetuned models—all while yielding a reusable rule library that transfers across LLMs.

While simple in spirit, this paradigm offers a step toward more modular and transparent LLM adaptation. By learning from failures and encoding recoverable patterns as structured rules, we move closer to systems that adapt in a more human-like way: abstracting from experience, reusing learned knowledge, and doing so in a form that is interpretable and composable. We hope this work encourages further exploration of inference-time abstraction as a foundation for reliable, transparent, and reusable augmentation of language agents.

REPRODUCIBILITY STATEMENT

We have provided the implementation details for reproducing the method and experiments. All datasets used in this paper are publicly available, and we present prompts in Appendix A.3, and pseudo code implementation in Appendix A.1. All the code will be released upon acceptance.

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

# A  APPENDIX

## A.1  PSEUDOCODE FOR RIMRULE

---

**Algorithm 1** RIMRULE: MDL-Guided Neuro-Symbolic Adaptation via Rules

---

**Require:** Training data $D = \{(x_i, \mathcal{S}_i, \tau_i^-, \tau_i^*)\}_{i=1}^n$ with failure traces
**Ensure:** Compact, symbolic rule library $\mathcal{R}$

    **Stage 1: Distributed Rule Generation**
 1: **for all** $(x_i, \mathcal{S}_i, \tau_i^-, \tau_i^*) \in D$ **in parallel do**
 2:    $r_i^{\mathrm{NL}} \leftarrow$ LLM-Propose-Rule$(x_i, \mathcal{S}_i, \tau_i^-, \tau_i^*)$
 3:    **if** Passes Predictive and Linguistic Checks **then**
 4:        Add $r_i^{\mathrm{NL}}$ to candidate rule pool $\mathcal{R}_0$
 5:    **end if**
 6: **end for**
    **Symbolic Translation**
 7: Build field-wise vocabulary $\mathcal{V}$ from $\mathcal{R}_0$
 8: **for all** $r_i^{\mathrm{NL}} \in \mathcal{R}_0$ **do**
 9:    $r_i^{\mathrm{can}} \leftarrow$ TranslateToSymbolic$(r_i^{\mathrm{NL}}, \mathcal{V})$
10: **end for**
11: Initialize $\mathcal{R} \leftarrow \mathcal{R}_0^{\mathrm{can}}$
    **Stage 2: MDL-Guided Consolidation**
12: **repeat**
13:    **for all** $r \in \mathcal{R}$ **do**
14:        Evaluate PRUNE$(r)$: $\Delta L_{\mathrm{MDL}}$
15:        Evaluate GENERALIZE$(r)$: $\Delta L_{\mathrm{MDL}}$
16:        **if** Best $\Delta L_{\mathrm{MDL}} < 0$ **then**
17:            Apply best edit to update $\mathcal{R}$
18:        **end if**
19:    **end for**
20: **until** no edit reduces MDL objective
21: **return** $\mathcal{R}$

---

## A.2  EXPERIMENT DETAILS

Table 6 summarizes the dataset splits used for training and evaluation, including the number of examples per split. For each dataset, we report performance on both **test-rand** (random split) and **test-unseen** (held-out tools).

Table 6: Number of examples in each dataset split. **test-unseen** is constructed by selecting queries whose tools are not seen during training.

| Dataset | Train | Test-rand | Test-unseen |
|---|---|---|---|
| ToolHop | 392 | 70 | 51 |
| BFCL:Live-Multiple | 735 | 175 | 143 |
| BFCL:Multi-Turn-Base | 90 | 60 | 50 |

Table 7 lists the full names and sources of the LLMs evaluated throughout the paper. For brevity, we refer to each model by a shortened name in tables and figures.

## A.3  PROMPTS

This section includes the prompts used throughout our pipeline: for generating rules from failure traces, discovering and updating the classification vocabulary, and translating rules into canonical form. Prompts are shown as used, with placeholders for query, tools, and traces.

Table 7: Short names used for LLMs in evaluation tables.

| Short Name | Full Model Name |
|---|---|
| Llama3.2 | `meta.llama3-2-3b-instruct-v1-0` |
| Llama4 | `meta.llama4-maverick-17b-instruct-v1-0` |
| GPT-4o | `gpt-4o-2024-11-20` |
| O1 | `o1-mini-2024-09-12` |

```
You are a senior AI systems instructor tasked with helping a tool-using
    language model, referred to as the Tool Agent, improve its ability
    to use tools accurately, efficiently, and reliably, even with new
    tools and unseen queries that are structurally and/or semantically
    similar to previously seen examples. Your role is to review
    incorrect tool usage traces produced by the Tool Agent, compare them
    against the correct (groundtruth) traces, and analyze any embedded
    error messages to reason about the error.

An incorrect trace occurs when a query's execution by the Tool Agent
    doesn't match the groundtruth answer. Incorrect traces are either
    complete (finished but incorrect) or incomplete due to unrecoverable
    execution errors. You will be provided with the Tool Agent's full
    execution trace, including embedded error messages.

There are two types of errors:
i. Reasoning errors (to analyze and generate rules for)
ii. Propagation errors (to ignore, as they result from prior mistakes)

Steps may contain:
(i) no errors, (ii) only propagation errors, or (iii) one or more
    reasoning errors.

Your goal is to extract generalizable, tool-schema-aware, and
    context-sensitive rules that can guide future tool usage-even with
    unseen queries and schemas. Think like a meta-cognitive teacher:
    diagnose root causes and articulate precise, reusable abstractions.

You will now be given:
1. The user query [INSERT USER QUERY]
2. The available tools [INSERT TOOL SCHEMA JSON]
3. The incorrect trace [INSERT TOOL AGENT TRACE]
4. The groundtruth trace [INSERT GROUNDTRUTH TRACE]

Your task is to identify reasoning errors by comparing the traces.
Check:
- Were subtasks identified correctly?
- Were the correct tools selected?
- Were the arguments constructed correctly?

Guidelines for rule generation:
- Only generate rules for reasoning errors
- Identify root causes, not superficial fixes
- Rules must be generalizable and schema-aware
- Rules must not contain query-specific or tool-specific tokens
- Each rule must be atomic and classifiable as:
 a. decomposition error
 b. tool selection error
 c. tool arguments error
- If multiple reasoning failures occur, generate multiple rules unless
    clarity is preserved
```

```
Rules must be symbolic, composable, and conform to a standard form
    (e.g., condition => action).

Output format:
First, write a brief analysis explaining the root cause and error type.
Then output one rule only, using this JSON structure:
{
  "new_rule": "<generalized if-then rule>",
  "error_type": "<decomposition error / tool selection error / tool
      arguments error>"
}
```

Listing 1: Prompt used to generate reasoning rules from failure traces. Placeholders like user query and tool schemas are injected at runtime.

```
You are an expert rule classification specialist.
Your task is to CREATE the initial vocabulary for rule classification
    based on the provided field definitions.
Analyze the rules carefully and create comprehensive categories that
    cover all rule types.
Use only uppercase letters, numbers, and underscores for token names.
Return only valid JSON format.

Create initial vocabulary for rule classification based on these
    definitions:

FIELD DEFINITIONS:

DOMAIN (enum): Broad topical domain, inferred jointly from rule content
    + tool context + query context.

QUALIFIER (enum): Fine-grained situational tags that are more specific
    than domain.

ACTION (enum): The action(s) the rule prescribes, from a closed set.

STRENGTH (enum): Rule action priority (must, may). Same
    scope/domain/qualifier, higher priority should be retrieved rather
    than low priority.
Examples: MANDATORY, RECOMMENDED, OPTIONAL

TOOL_CATEGORY (enum): Tool functional category based on tool
    capabilities and usage patterns.
This should represent broad tool types that can group similar tools
    together.
Examples: DATA_PROCESSING, SEARCH_ENGINE, COMPUTATION, TEXT_PROCESSING...

Rules to analyze:
[INSERT_BULLETS_HERE]

Create a comprehensive vocabulary covering all rule types in the above
    rules.
Use only uppercase letters, numbers, and underscores for category names.

Return JSON:
{
  "vocab_version": "v1",
  "domain": [ ... ],
  "qualifier": [ ... ],
  "action": [ ... ],
  "strength": [ ... ],
  "tool_category": [ ... ]
}
```

Listing 2: Prompt used to induce the rule classification vocabulary from a batch of rules. Placeholders are filled dynamically.

```
You are an expert rule classification specialist.
Your task is to UPDATE and EXPAND an existing vocabulary based on new
    rules.
Review the current vocabulary, identify missing categories from new
    rules, and return the complete updated vocabulary.
Use only uppercase letters, numbers, and underscores for token names.
Avoid duplicates and maintain consistency with existing categories.
Return only valid JSON format.

Update and expand vocabulary for rule classification.

FIELD DEFINITIONS:

DOMAIN (enum): Broad topical domain, inferred jointly from rule content
    + tool context + query context.

QUALIFIER (enum): Fine-grained situational tags that are more specific
    than domain.

ACTION (enum): The action(s) the rule prescribes, from a closed set.

STRENGTH (enum): Rule action priority (must, may). Same
    scope/domain/qualifier, higher priority should be retrieved rather
    than low priority.

TOOL_CATEGORY (enum): Tool functional category based on tool
    capabilities and usage patterns.

Current accumulated vocabulary:
DOMAIN: [INSERT CURRENT DOMAINS OR 'None yet']
QUALIFIER: [INSERT CURRENT QUALIFIERS OR 'None yet']
ACTION: [INSERT CURRENT ACTIONS OR 'None yet']
STRENGTH: [INSERT CURRENT STRENGTHS OR 'None yet']
TOOL_CATEGORY: [INSERT CURRENT TOOL_CATEGORIES OR 'None yet']

New rules to analyze:
[INSERT NEW RULE BULLETS HERE]

Instructions:
1. Review the current vocabulary above
2. Analyze the new rules to identify any missing categories
3. Add new categories if needed, but avoid duplicates
4. Keep existing categories that are still relevant
5. Return the complete updated vocabulary

Return JSON with all categories (existing + new):
{
  "domain": ["existing_domains", "new_domains_if_any"],
  "qualifier": ["existing_qualifiers", "new_qualifiers_if_any"],
  "action": ["existing_actions", "new_actions_if_any"],
  "strength": ["existing_strengths", "new_strengths_if_any"],
  "tool_category": ["existing_tool_categories",
      "new_tool_categories_if_any"]
}
```

Listing 3: Prompt used to update and expand the rule classification vocabulary using newly generated rules.

```
You are an expert rule classification specialist.
```

```
Your task is to classify individual rules using the provided vocabulary.
Analyze the rule content, tool context, and query context to determine
    the most appropriate classification.
Choose only from the given vocabulary options.
Return only valid JSON format.

Vocab enumerations:
- domain: [INSERT domain list]
- qualifier: [INSERT qualifier list]
- action: [INSERT action list]
- strength: [INSERT strength list]
- tool_category: [INSERT tool_category list]

Rule to classify (analyze all fields for context):
[INSERT RULE OBJECT AS JSON]

Classification Guidelines:

DOMAIN: Infer from rule content + tool context + query context. Choose
    the broadest applicable domain that covers the rule's topic.

QUALIFIER: Select fine-grained situational tags that are more specific
    than domain. You can select multiple qualifiers if the rule applies
    to multiple situations. Focus on conditions, constraints, or
    specific contexts mentioned in the rule.

ACTION: Identify the specific action(s) the rule prescribes. Choose from
    the closed set of available actions. You can select multiple actions
    if the rule prescribes multiple steps.

STRENGTH: Determine rule priority (must, may). MANDATORY = must follow,
    RECOMMENDED = should follow, OPTIONAL = may follow. Consider the
    rule's language and context to determine priority.

TOOL_CATEGORY: Identify the functional category of the tool(s) used in
    this rule. Choose from the available tool categories based on the
    tool's capabilities and usage pattern. This should represent the
    broad functional type of the tool (e.g., DATA_PROCESSING,
    SEARCH_ENGINE, COMPUTATION).

Return ONLY valid JSON matching this schema:
{
  "_id": 0,
  "domain": "FAMILIAL_RELATIONSHIPS",
  "qualifier": ["SEQUENTIAL", "MULTI_STEP"],
  "action": ["DECOMPOSE", "VALIDATE"],
  "strength": "MANDATORY",
  "tool_category": "DATA_PROCESSING"
}
```

Listing 4: Prompt used to classify individual rules using a fixed vocabulary. Vocabulary values and rule content are injected at runtime.

### A.4    SAMPLE RULES

Below, we present a representative subset of learned rules. Each rule is shown in natural language along with its corresponding symbolic representation.

```
1. If the user query involves identifying a complex relationship (e.g.,
    step-relative) and the available tools do not directly support the
    requested relationship type, then decompose the query into
    intermediate subtasks that progressively resolve the relationship
    through simpler, directly supported relationship types.
```

```
if (domain=RELATIONSHIP_RESOLUTION and
    qualifier=[RELATIONSHIP_CHAIN_TRAVERSAL,
    INTERMEDIATE_ENTITY_IDENTIFICATION]) then (action=[DECOMPOSE_QUERY,
    RESOLVE_INTERMEDIATE_ENTITY]) with strength=MANDATORY

2. If a query involves determining a property of an object derived from
   intermediate steps (e.g., counting, analyzing, or transforming an
   attribute of an entity), then the decomposition must explicitly
   include a subtask to apply the appropriate process or transformation
   to the derived attribute.
if (domain=QUERY_DECOMPOSITION and qualifier=[MULTI_STEP_REASONING,
    DERIVATIVE_ENTITY]) then (action=[DECOMPOSE_QUERY, TRANSFORM_INPUT])
    with strength=MANDATORY

3. If a query fails due to insufficient or incomplete data returned by a
   tool, then refine the input arguments by including optional
   parameters that provide additional context (e.g., variants of names,
   time periods, regions, or other applicable constraints) to improve
   data retrieval accuracy.
if (domain=TOOL_ARGUMENT_VALIDATION and
    qualifier=[INCOMPLETE_DATA_REFINEMENT]) then
    (action=[REFINE_INPUT_ARGUMENTS, HANDLE_INCOMPLETE_DATA]) with
    strength=RECOMMENDED

4. If a query involves determining an attribute (e.g., date, location,
   event) of an entity that is related to another entity (e.g., family
   member), then decompose the query to first identify the related
   entity before attempting to determine its attribute.
if (domain=QUERY_DECOMPOSITION and
    qualifier=[RELATIONSHIP_CHAIN_TRAVERSAL,
    INTERMEDIATE_ENTITY_IDENTIFICATION]) then (action=[DECOMPOSE_QUERY,
    RESOLVE_INTERMEDIATE_ENTITY]) with strength=MANDATORY

5. If the subtask requires extracting a specific subset of information
   (e.g., first name from a full name), then construct arguments for
   the tool to restrict the output to match the requested subset by
   enabling relevant optional parameters explicitly, while leaving
   unrelated parameters at their defaults.
if (domain=NAME_PROCESSING and qualifier=[NAME_COMPONENT_EXTRACTION])
    then (action=[CONSTRUCT_ARGUMENTS, REFINE_INPUT_ARGUMENTS]) with
    strength=MANDATORY

6. If the user query involves identifying information about a specific
   familial relationship across multiple generations (e.g., paternal
   grandfather), then decompose the query such that the familial
   relationship is resolved directly using tools designed to retrieve
   genealogical data, without redundantly extracting unrelated
   immediate relationships.
if (domain=RELATIONSHIP_RESOLUTION and qualifier=[FAMILIAL_RELATIONSHIP,
    RELATIONSHIP_CHAIN_TRAVERSAL]) then (action=[DECOMPOSE_QUERY,
    IDENTIFY_RELATIONSHIP]) with strength=MANDATORY

7. If a query involves determining the ancestry of a person based on a
   specific familial relationship (e.g., maternal grandfather), then
   decompose the task into subtasks that sequentially query each
   familial link (e.g., mother, mother's father) and ensure
   intermediate results are used to refine subsequent subtasks.
if (domain=HIERARCHICAL_RELATIONSHIPS and
    qualifier=[FAMILIAL_RELATIONSHIP, RELATIONSHIP_CHAIN_TRAVERSAL,
    INTERMEDIATE_ENTITY_IDENTIFICATION]) then (action=[DECOMPOSE_QUERY,
    SEQUENCE_SUBTASKS, RESOLVE_INTERMEDIATE_ENTITY]) with
    strength=MANDATORY

8. If a query involves identifying an entity indirectly related to
   another (e.g., the founder of a political party identified in a
```

```
      previous step), then ensure that intermediate subtasks chain outputs
      logically by querying the relationship explicitly rather than
      assuming associations provided in the query.
if (domain=INDIRECT_ENTITY_IDENTIFICATION and
      qualifier=[RELATIONSHIP_CHAIN_TRAVERSAL,
      INTERMEDIATE_ENTITY_IDENTIFICATION]) then
      (action=[SEQUENCE_SUBTASKS, RESOLVE_INTERMEDIATE_ENTITY]) with
      strength=MANDATORY

9. If a query involves retrieving information about an individual's
      relative, the decomposition must include a subtask to explicitly
      identify the relative before retrieving specific information about
      them.
if (domain=RELATIONSHIP_RESOLUTION and qualifier=[FAMILIAL_RELATIONSHIP,
      RELATIONSHIP_CHAIN_TRAVERSAL]) then (action=[DECOMPOSE_QUERY,
      RESOLVE_INTERMEDIATE_ENTITY]) with strength=MANDATORY

10. If the subtask involves identifying a familial or social
      relationship between entities, then select a tool designed to query
      relationships, and do not select a tool designed for extracting name
      components or parsing names.
if (domain=RELATIONSHIP_RESOLUTION and qualifier=[FAMILIAL_RELATIONSHIP,
      RELATIONSHIP_CHAIN_TRAVERSAL]) then (action=[MATCH_TOOL_TO_SUBTASK])
      with strength=MANDATORY
```

Listing 5: Sample rules learned from the ToolHop dataset

```
1. If a reasoning process identifies a subtask requiring tool-based
      execution, then ensure the tool name generated in the output matches
      exactly with one of the available tools in the schema. Use schema
      validation to confirm that the generated tool name exists before
      finalizing the response.
if (domain=TRANSPORTATION and qualifier=[SCHEMA_VALIDATION,
      TOOL_SELECTION_ERROR]) then (action=[CONFIRM_SCHEMA_COMPLIANCE,
      SELECT_TOOL_BASED_ON_QUERY]) with strength=MANDATORY

2. IF the query involves retrieving general information about an entity,
      concept, or event that is not explicitly tied to recent updates or
      temporal relevance, THEN select a tool designed for general web
      searches instead of a tool specialized for retrieving recent news or
      updates.
if (domain=GENERAL_INFORMATION_RETRIEVAL and
      qualifier=[TOOL_SELECTION_ERROR]) then
      (action=[SELECT_TOOL_BASED_ON_QUERY]) with strength=MANDATORY

3. If the subtask requires checking availability or stock levels for
      specific product attributes (e.g., sizes, stock count), then select
      a tool whose schema explicitly supports inventory-related
      operations, rather than using a tool designed for general product
      details.
if (domain=INVENTORY_MANAGEMENT and qualifier=[TOOL_SELECTION_ERROR])
      then (action=[SELECT_TOOL_BASED_ON_QUERY]) with strength=MANDATORY

4. If the subtask involves checking the availability of a specific
      attribute (e.g., size, color, or stock) for a product, then select a
      tool whose schema explicitly includes functionality for querying
      availability of attributes rather than retrieving general product
      details.
if (domain=INVENTORY_MANAGEMENT and qualifier=[TOOL_SELECTION_ERROR])
      then (action=[SELECT_TOOL_BASED_ON_QUERY]) with strength=MANDATORY

5. If a user query explicitly provides a value for a parameter,
      regardless of whether the parameter has a default value in the
```

```
1080    schema, prioritize the user-provided value over the default value
1081    when constructing arguments.
1082  if (domain=DATABASE_MANAGEMENT and qualifier=[USER_QUERY_VALUE_PRIORITY,
1083    DEFAULT_VALUE_ASSIGNMENT]) then (action=[PRIORITIZE_USER_VALUE,
1084    CONSTRUCT_TOOL_ARGUMENTS]) with strength=MANDATORY
1085
1086  6. If the user query explicitly requests understanding, explanation, or
1087    help about a functionality, then select a tool designed to provide
1088    informational assistance rather than one intended for operational
1089    execution.
      if (domain=GENERAL_INFORMATION_RETRIEVAL and
1090    qualifier=[TOOL_SELECTION_ERROR]) then
1091    (action=[SELECT_TOOL_BASED_ON_QUERY]) with strength=MANDATORY

1092  7. If the intended action is to retrieve information based on
1093    user-specified filters or constraints, then select a tool that
1094    explicitly supports filtering or constraint-based retrieval in its
1095    schema description. Ensure the subtask aligns directly with the
1096    tool's primary function as specified in its schema.
      if (domain=GENERAL_INFORMATION_RETRIEVAL and
1097    qualifier=[TOOL_SELECTION_ERROR]) then
1098    (action=[SELECT_TOOL_BASED_ON_QUERY]) with strength=MANDATORY
1099
1100  8. If a subtask involves identifying available options or exploring
1101    possible choices for a resource (e.g., accommodations, flights,
1102    attractions), then select a tool designed for searching or browsing
1103    those resources rather than a tool designed for finalizing or
      reserving them.
1104  if (domain=GENERAL_INFORMATION_RETRIEVAL and
1105    qualifier=[TOOL_SELECTION_ERROR]) then
      (action=[SELECT_TOOL_BASED_ON_QUERY]) with strength=MANDATORY
1106
1107  9. If the user query specifies a constraint related to an entity (e.g.,
1108    actor name, director name, genre, etc.), then construct the
1109    corresponding parameter value by directly extracting the entity
1110    mentioned in the query, instead of using default or placeholder
      values.
1111  if (domain=GENERAL_INFORMATION_RETRIEVAL and
1112    qualifier=[REQUIRED_PARAMETER_FORMATTING,
1113    USER_QUERY_VALUE_PRIORITY]) then (action=[CONSTRUCT_TOOL_ARGUMENTS,
1114    PRIORITIZE_USER_VALUE]) with strength=MANDATORY
1115
1116  10. If a schema explicitly limits valid values for an input parameter to
1117    a predefined set, then when constructing arguments, map user
1118    preferences to the closest valid value within this set, or default
1119    to a neutral option (e.g., no filtering) when the schema disallows
      explicit preferences.
1120  if (domain=GENERAL_INFORMATION_RETRIEVAL and
1121    qualifier=[SCHEMA_VALIDATION, DEFAULT_VALUE_ASSIGNMENT]) then
1122    (action=[CONSTRUCT_TOOL_ARGUMENTS, ASSIGN_DEFAULT_VALUE]) with
      strength=MANDATORY
1123
```

Listing 6: Sample rules learned from the BFCL: Live-Multiple dataset

```
1126  1. If a subtask involves initiating a system or process with
1127    preconditions, then identify and include all required preconditions
1128    (e.g., state validations, sequential actions) in the task
1129    decomposition, ensuring that they are executed in the correct order
1130    prior to invoking the main action. For example, if starting a
1131    process requires preconditions A, B, and C, verify and satisfy A, B,
      and C sequentially before initiating the process.
1132  if (domain=PROCESS_MANAGEMENT and qualifier=[PRECONDITION_VALIDATION,
1133    TASK_SEQUENCE_VALIDATION]) then (action=[DECOMPOSE_TASK,
      VALIDATE_PRECONDITIONS, ENSURE_SEQUENCE]) with strength=MANDATORY
```

```
2. If a user query involves an action that requires specific
   prerequisites to be met (e.g., a dependency between tasks or a state
   requirement), decompose the query into subtasks that explicitly
   address the prerequisites first. Ensure each prerequisite subtask is
   executed and validated before proceeding with the dependent action.
if (domain=PROCESS_MANAGEMENT and qualifier=[PREREQUISITE_TASK,
   DEPENDENCY_MANAGEMENT, TASK_SEQUENCE_VALIDATION]) then
   (action=[DECOMPOSE_TASK, VALIDATE_PRECONDITIONS, EXECUTE_SUBTASK])
   with strength=MANDATORY

3. If the task involves converting a monetary target between two
   currencies, then ensure the base currency matches the source of the
   monetary target and the target currency matches the destination
   specified in the query context. Do not reverse these roles, as it
   will misalign the output with the intended goal.
if (domain=CURRENCY_CONVERSION and qualifier=[CURRENCY_CONSISTENCY,
   PRECONDITION_VALIDATION]) then (action=[MATCH_CURRENCY,
   VALIDATE_PRECONDITIONS]) with strength=MANDATORY

4. If a query requires identifying entities from a complete set (e.g.,
   all available options), then retrieve the comprehensive set
   explicitly using tools designed for global enumeration before
   attempting subtasks that depend on specific entities. Do not attempt
   to infer the set by piecemeal or localized retrieval from individual
   components.
if (domain=ENTITY_RETRIEVAL and qualifier=[GLOBAL_ENUMERATION,
   DEPENDENCY_MANAGEMENT]) then (action=[RETRIEVE_GLOBAL_SET,
   EXECUTE_SUBTASK]) with strength=MANDATORY
```

Listing 7: Sample rules learned from the BFCL: Multi-Turn-Base dataset

## A.5 USE OF LLMS

ChatGPT[4] is used to polish the writing of this paper.

---

[4] https://chatgpt.com