# OpenReview forum: "Improving Tool-Using Language Agents via MDL‑Guided Rule Learning"
_ICLR.cc/2026/Conference — ICLR 2026 Conference Withdrawn Submission_

### Official Review · Reviewer_q37q · 2025-10-31

**Soundness:** 2
**Presentation:** 3
**Contribution:** 2
**Rating:** 4
**Confidence:** 3

**Summary:**

This paper proposes RimRule, a method that improves how language models use tools by learning interpretable rules from their mistakes. Instead of retraining the model, it generates rules from failure traces, compresses them into a concise library using a Minimum Description Length objective, and injects the most relevant rules into the model's prompt during inference. Experiments show this approach boosts performance on both seen and unseen tools, outperforms other prompting techniques, complements finetuning, and allows rules learned by one model to successfully improve others.

**Strengths:**

1. This paper addresses an important topic—how large language models can efficiently learn from experience. The proposed method outperforms supervised fine-tuning, demonstrating strong potential for broader applications.
2. The study tackles a significant challenge: efficient management of an agent’s experiential knowledge. The approach is novel and conceptually promising, though several concerns remain.

**Weaknesses:**

1. The experimental pipeline relies heavily on manually designed priors—including rule structures, retrieval methods, and optimization strategies—raising concerns about generalizability and scalability.
2. The experimental evaluation appears incomplete:
3. The rationale for dataset selection is unclear.
4. The baselines are overly simple; comparable works (e.g., AWM) are not included. The choice of LoRA for Llama 3.2-3B is questionable, as full fine-tuning seems computationally feasible. Models from the Qwen family are also omitted. No case studies are provided.

**Questions:**

1. The use of the MDL-based rule pruning formula may be problematic. While the decomposition
$(L(H) + L(D \mid H))$  is valid, the paper computes the two terms inconsistently: $L(H)$ is approximated by the natural language description length, which is not an optimal semantic encoding. The determination of the coefficient $\alpha$ is also unexplained, despite its direct influence on pruning decisions.
2. The goal of rule pruning should be performance improvement, not merely computational efficiency. However, minimizing $L(D \mid H)$ does not inherently favor performance gains—it can be minimized whether all cases succeed or fail. Thus, the outcome likely depends on the initial success rate: if it exceeds 0.5, the model is biased toward success; otherwise, toward failure.

---

### Official Review · Reviewer_ZAir · 2025-10-31

**Soundness:** 2
**Presentation:** 2
**Contribution:** 2
**Rating:** 2
**Confidence:** 3

**Summary:**

The authors propose a method to extract symbolic rules from execution traces which either fail or are incomplete. These rules can be injected in the prompt at inference time and they show that they increase performance compared to traditional few shot prompting and other baselines. Furthermore, the set of rules can be reduced by leveraging the Minimum Description Length. The authors evaluate their approach using two tool use benchmarks: ToolHop and BFCL. They compare it against vanilla prompting-based methods as well as a prompt optimization baseline (SEE). They also showcase that it still brings a performance boost even when the model has been finetuned for tool calling - although no details are provided about which data was used for this SFT step.

**Strengths:**

- The authors present a method that benefits tool calling at inference time for models that have not been finetuned for using tools. Specifically, they show case that their method helps for unseen tools.

**Weaknesses:**

- In the Rule retrieval section, it is confusing that naively using retrieval with an embedding model for natural language and a semantic similarity criterion is discouraged but they you still propose to use an embedding model for NLP data not tailored for symbolic rules). Ideally, one should train an embedding model for rules and not use semantic similarity but a measure of similarity that makes more sense for the space of rules.
- The word “training” is used sometimes without enough context, for instance, “the rule library whose size and content evolves during training” but my understanding is that this training only refers to using each incorrect/incomplete trace to produce a rule since no objective function is presented except for the one used in the rule consolidation section. However, I assume the objective function in the MDL-guided rule consolidation section is only for the consolidation step.
 - Missing baselines wrt other symbolic rule generation methods.
- The paper organization could be improved since it is hard to read at times.

**Questions:**

- It is assumed that you have access to execution traces and incorrect or incomplete execution traces in particular. What is the number of failed traces required to obtain a boost in performance?
- The term pre-fine tuned is used in section 3.2, what does it refer to?
- Is it possible to conduct an ablation where you generate the rules with two models e.g. 8B and 70B and then evaluate these wrt the benchmarks? Even when the paper presents results that showcase that the rules produced by one model can be reused at inference time by any other model, it is unclear if there is a boost when you use a more powerful model to generate the rules
- There are no details provided on how the SFT is done, what data did you use for SFT presented in table 4?
- As mentioned in the weaknesses, I believe some baselines are missing, e.g. other symbolic rule generation methods. A very simple baseline that is also missing is to use the extracted rules in natural language and use a vanilla retrieval step with semantic similarity at inference time wrt the query

---

### Official Review · Reviewer_Uirv · 2025-11-01

**Soundness:** 3
**Presentation:** 2
**Contribution:** 2
**Rating:** 6
**Confidence:** 1

**Summary:**

This work proposed a neuro-symbolic method that generates a set of of tool use rules for unfamiliar tools in tool reasoning tasks. the method includes two phases: generating tool rules by comparing the predicted tool use trajectories against groundtruth traces. Second step is compressing the generated rules following the minimal description length rule.

Experiments show that applying the rules to the agents and further consolidating the generated rules both improves the performance of tool use performance significantly.

**Strengths:**

Experiment results support the main claims of the study pretty well. The generated rules can help with the tool use ability of language model-based agent, and consolidating rules can further improve the performance.

**Weaknesses:**

I'm not super familiar of neuro-symbolic methods, so I not confident about finding the weakness of the paper. The only thoughts I have is that the study could run more ablation studies to compare differnt rule consolidation strategies.

**Questions:**

see weakness

---

### Official Review · Reviewer_Jztf · 2025-11-01

**Soundness:** 2
**Presentation:** 2
**Contribution:** 2
**Rating:** 4
**Confidence:** 3

**Summary:**

This paper presents RIMRULE, a neuro-symbolic approach for improving LLM tool-use capabilities through dynamic rule injection. The method learns compact, interpretable rules from failure traces, consolidates them using Minimum Description Length (MDL) principles, and retrieves relevant rules at inference time to guide agent behavior. Experiments on ToolHop and BFCL demonstrate improvements over prompting-based methods and complementarity with fine-tuning.

**Strengths:**

The paper introduces a fourth paradigm for LLM adaptation—symbolic rule learning—that complements existing approaches (few-shot prompting, prompt optimization, fine-tuning).

**Weaknesses:**

The paper positions itself as addressing tool-use adaptation but does not engage with recent work on unified approaches to tool retrieval and calling. In particular, they should discuss work on unified tool retrieval and calling via generation, explaining how their symbolic approach offers different trade-offs (interpretability, modularity) compared to end-to-end generation methods.

The greedy MDL optimization (Section 2.4) raises scalability questions:
How does consolidation time scale with the number of candidate rules?
How far is the greedy solution from optimal? Have the authors explored approximate inference methods?
The α parameter controls the length penalty—how sensitive are results to this choice? An ablation study is needed.

The paper reports that consolidation reduces rules (e.g., 151→121 for BFCL) but doesn't quantify the computational cost or approximation quality.

**Questions:**

While the paper reports aggregate accuracy improvements, it provides limited insight into:
What makes a rule "good"? Beyond MDL metrics, what semantic properties characterize effective rules?
When do rules help vs. hurt? Are there cases where injected rules degrade performance? What proportion of failures are addressed by learned rules? How many failures remain unaddressed?

---

### Note · Authors · 2025-11-26

I have read and agree with the venue's withdrawal policy on behalf of myself and my co-authors.